# Alcohol Exposure and Disease Associations: A Mendelian Randomization and Meta-Analysis on Weekly Consumption and Problematic Drinking

**DOI:** 10.3390/nu16101517

**Published:** 2024-05-17

**Authors:** Mengyao Li, Xuying Zhang, Kailei Chen, Yang Miao, Yaxin Xu, Yishuo Sun, Mengxian Jiang, Mengcao Liu, Yan Gao, Xiaoxia Xue, Xuelian Li

**Affiliations:** 1Key Laboratory of Environmental Stress and Chronic Disease Control & Prevention, China Medical University, Ministry of Education, Shenyang 110122, China; 15308159279@163.com (M.L.); 2022120052@cmu.edu.cn (X.Z.); 2022120128@cmu.edu.cn (K.C.); 2023120041@cmu.edu.cn (Y.M.); 2023120043@cmu.edu.cn (Y.X.); yssun0823@163.com (Y.S.); jiangmengxian0605@163.com (M.J.); liumengcao0727@163.com (M.L.); gaoyan152@yeah.net (Y.G.); 2Department of Epidemiology, School of Public Health, China Medical University, Shenyang 110122, China; 3Science Experiment Center, China Medical University, Shenyang 110122, China; xxxue@cmu.edu.cn

**Keywords:** alcohol consumption, problematic alcohol use, Mendelian randomization analysis, meta-analysis, disease risk

## Abstract

Alcohol consumption significantly impacts disease burden and has been linked to various diseases in observational studies. However, comprehensive meta-analyses using Mendelian randomization (MR) to examine drinking patterns are limited. We aimed to evaluate the health risks of alcohol use by integrating findings from MR studies. A thorough search was conducted for MR studies focused on alcohol exposure. We utilized two sets of instrumental variables—alcohol consumption and problematic alcohol use—and summary statistics from the FinnGen consortium R9 release to perform de novo MR analyses. Our meta-analysis encompassed 64 published and 151 de novo MR analyses across 76 distinct primary outcomes. Results show that a genetic predisposition to alcohol consumption, independent of smoking, significantly correlates with a decreased risk of Parkinson’s disease, prostate hyperplasia, and rheumatoid arthritis. It was also associated with an increased risk of chronic pancreatitis, colorectal cancer, and head and neck cancers. Additionally, a genetic predisposition to problematic alcohol use is strongly associated with increased risks of alcoholic liver disease, cirrhosis, both acute and chronic pancreatitis, and pneumonia. Evidence from our MR study supports the notion that alcohol consumption and problematic alcohol use are causally associated with a range of diseases, predominantly by increasing the risk.

## 1. Introduction

Alcohol, a psychoactive substance with dependence-producing properties, has been widely used in many cultures for centuries [1]. Alcohol use ranks seventh among risk factors contributing to the burden of disease and results in significant health loss. The overall impact of alcohol consumption on health is negative, accounting for approximately 3.8% of all global deaths and 4.6% of global disability-adjusted life-years attributable to alcohol [2]. Numerous diseases, including ischemic conditions and diabetes, show a consistent relationship with the volume of alcohol consumed [3]. However, traditional observational studies may be affected by environmental confounding and reverse causation bias.

Mendelian randomization (MR) is an analytical method that utilizes genetic variants associated with a modifiable exposure or biological intermediaries as instrumental variables (IVs). This approach is employed to infer causal relationships between these variables and health-related outcomes, effectively mitigating confounding and reducing bias from reverse causation [4]. Recent advancements in Genome-Wide Association Studies (GWASs) have identified numerous Single Nucleotide Polymorphisms (SNPs) linked to various drinking behaviors, including alcohol consumption (quantified as “drinks per week”) and “problematic alcohol use” [5,6]. MR has been increasingly utilized to explore the causal links between alcohol consumption and disease susceptibility. However, comprehensive reviews and meta-analyses that utilize Mendelian randomization (MR) to examine drinking patterns remain scarce.

To comprehensively investigate and synthesize evidence on the causal impact of alcohol use across various diseases, we conducted a systematic literature search of published MR analyses that focus on diseases associated with alcohol consumption risks. We utilized two sets of instrumental variables—alcohol consumption (measured in “drinks per week”) and problematic alcohol use. We used summary statistics from Genome-Wide Association Studies (GWASs), which included data from the FinnGen study [7] and other publicly accessible GWAS datasets, to conduct de novo Mendelian randomization (de novo MR) analyses [6]. Ultimately, we performed meta-analyses to aggregate and critically assess the results derived from both published MR analyses and our de novo MR analyses. Our study aims to summarize the results of Mendelian randomization (MR) studies on genetic predictors of drinking patterns, providing evidence-based assessments of health risks associated with alcohol consumption.

## 2. Materials and Methods

### 2.1. Literature Search and Inclusion Criteria

Four databases, Pubmed, Embase, Web of Science, and Scopus, were searched for relevant MR analyses published until 6 March 2023, employing the terms “alcohol consumption” and “GWAS” (Appendix A). The scope of our research included original full-text articles examining the association between alcohol consumption (drink per week) or problematic alcohol use (pathological drinking behaviors) and a wide spectrum of disorders: circulatory, digestive, neurological, respiratory, genitourinary, and musculoskeletal disorders, infections, diseases in pregnancy and puerperium, eye diseases, skin diseases, and tumors. Our study encompassed original articles from two sample MRs that were inclusive of all genders, ages, and races, stipulating the provision of accessible full text and the necessary data for our analyses. The exposure phenotypes considered were alcohol consumption or problematic alcohol use, without restrictions on sample size. We excluded studies utilizing only a single or a few genetic instrumental variables (<10) due to their limited scope and variability in exposure units. Outcome measures were aligned with the International Classification of Diseases, 10th edition (ICD10). In cases where articles originated from the same cohort data source, preference was given to those with larger sample sizes. For pooled data from identical GWAS sources, we selected studies based on higher quality scores, as detailed in Appendix A. The study adhered to the PRISMA guidelines [8] and was duly registered in PROSPERO (ID: CRD42023408294).

### 2.2. Data Extraction and Quality Control

All documents retrieved were imported into Rayyan [9] “http://rayyan.qcri.org (accessed on 16 March 2023)” for initial literature screening, with the specific screening criteria depicted in Appendix A in the Supplementary files. Given the absence of a standardized quality assessment tool for systematic reviews of MR analyses, we formulated a literature quality rating scale (Appendix A). This scale was developed by referencing the “Strengthening the Reporting of Observational Studies in Epidemiology Using Mendelian Randomization (STROBE-MR)” guidelines [10] and the literature quality assessment criteria employed in Luo et al.’s systematic review of MR [11]. Each piece of the included literature was independently evaluated and scored by two researchers (X.Z. and K.C.) based on those criteria. Discrepancies in scores were resolved by consulting a third researcher (M.L. (Mengyao Li)) for a definitive adjudication.

Data extraction from each study encompassed the primary author’s last name and publication year; PubMed unique identifier (PMID); exposure phenotype; consortium or study from which the genetic predisposition to the exposure phenotype was retrieved; outcome phenotype; consortium or study from which the genetic predisposition to the outcome phenotype was retrieved; ethnicity; sample size of outcome (i.e., number of cases and non-cases); number of IVs; and the relative risk estimate (odds ratio [OR]) with corresponding 95% confidence interval (CI) for the drinking-disease association from the main analysis based on the inverse-variance weighted (IVW) method as well as from sensitivity analysis based on weighed median (WME) and MR-Egger methods and multivariable MR analyses with adjustment for genetic predisposition to smoking initiation. Additionally, if the study underwent Mendelian Randomization Pleiotropy RESidual Sum and Outlier (MR-PRESSO), we collected the *p*-value of the global test and the number of outliers. The data extraction process was independently executed by two investigators (X.Z. and M.L. (Mengyao Li)) and rigorously cross-verified by another (X.L., X.X., and K.C.).

### 2.3. De Novo MR Analysis

The investigation into genetic determinants of drinking habits and smoking initiation incorporated GWAS studies on “alcohol consumption” and “smoking initiation” by Saunders et al., published in Nature [5], and Zhou et al.’s study on “problematic alcohol use”, featured in Nature Neuroscience [6]. In the GWAS studies, logarithmic changes in weekly intake were used to quantify alcohol consumption. And problematic alcohol use (PAU) was identified based on established diagnostic criteria: (a) alcohol use disorder diagnosed via ICD-10/9; (b) lifetime DSM-IV diagnosis of alcohol dependence; and (c) AUDIT-P scores. This research commenced with an initial screening of Single Nucleotide Polymorphisms (SNPs) associated with drinking phenotypes, adhering to a genome-wide association threshold of *p* < 5 × 10^−8^. Subsequently, linkage disequilibrium (LD) analyses were conducted on the identified SNPs using the Thousand Genomes Project Reference Panel of European populations. SNPs exhibiting an LD status (r^2^ > 0.1) were excluded from further consideration. F-statistics were calculated for each remaining SNP, and those exhibiting F < 10 were omitted. The final set of SNPs, having surpassed these stringent criteria, were then utilized as instrumental variables (IVs) for the drinking phenotype in the subsequent analyses.

The de novo MR analyses comprised 116 SNPs for alcohol consumption and 26 SNPs for problematic alcohol use as IVs (Appendix A). These IVs represent the genetic predisposition to alcohol consumption and problematic alcohol use, respectively.

In the de novo MR analysis of this study, comprehensive genetic data from the FinnGen consortium R9 release [7] were utilized as outcome variables. It is crucial to note that for the de novo MR analysis of severe confirmed respiratory infections due to COVID-19, data from version 7 of the COVID-19 Host Genetics Initiative (COVID-19 hg), published on 8 April 2022 [“https://www.covid19hg.org/results/r7/ (accessed on 19 October 2023)”], were employed. In the case of amyotrophic lateral sclerosis (ALS) de novo MR data, preference was given to the study by Iacoangeli et al., published in Cell Reports [12], over prior data sources, as detailed in Appendix A.

The MR-PRESSO global test was conducted to identify pleiotropy in IVs. Upon identifying pleiotropic SNPs, these were systematically removed one at a time through an outlier test. Subsequent de novo MR analyses were then executed using the IVW method post-elimination. Considering the genetic correlation between alcohol consumption and smoking initiation [13], adjustments were made using MVMR. Sensitivity analyses included both the MR-Egger and Weighted Median Estimator (WME) methods [14], and the heterogeneity of IVs effect estimates was evaluated using Cochran’s Q heterogeneity test (Q-test). The study was carried out in accordance with the STROBE-MR guidelines.

### 2.4. Meta-Analysis

This study incorporated a dual-source approach for meta-analysis, encompassing MR findings from the existing literature and our de novo MR analysis. We conducted meta-analyses on the MR association estimates derived from various documents in the literature as well as those obtained from our de novo MR. When the meta-analysis revealed heterogeneity, as indicated by the Q-test, we employed results from the random effects model. In cases of homogeneity, the fixed effects model results were utilized. The significance threshold, denoted by a two-sided 0.05 significance level, was adjusted using the Benjamini–Hochberg (B–H) corrected method. Notably, associations with a raw *p*-value below 0.05 but a B-H-adjusted *p*-value above 0.05 were categorized as suggestive. In contrast, associations with a B-H-adjusted *p*-value consistently below 0.05 were deemed statistically significant. This methodology ensured the statistical reliability and robustness of our findings.

All analyses and visualizations presented in this study were conducted using R (versions 4.3.1 and 4.3.2).

## 3. Results

### 3.1. Literature Search, Study Selection, and De Novo MR

A comprehensive search across multiple databases yielded 5054 articles, which included 580 from PubMed, 1348 from Embase, 1673 from Web of Science, and 1453 from Scopus. Of these, 31 articles that fulfilled the inclusion criteria were included in the study. The categorization of the included studies by outcome revealed the following: 14 articles related to circulatory system diseases [15,16,17,18,19,20,21], 27 addressing digestive system diseases [22,23,24,25,26,27], 1 concerning mental and behavioral disorders [28], 6 focusing on nervous system diseases [29,30,31,32,33,34], 26 on neoplasms [22,35,36,37,38,39], 11 on other diseases [33,40,41,42,43,44,45]. Furthermore, we extended to conducting 151 novel MR analyses, comprising 76 on alcohol consumption and 75 on problematic alcohol use. Additionally, a total of 76 distinct disease outcomes were analyzed, as detailed in Appendix A.

### 3.2. Study Description

The 31 studies incorporated in this research attained quality scores ranging from 7 to 12, signifying a literature quality at a moderate or higher level (Appendix A). Data on alcohol consumption for all incorporated studies were derived from the Alcohol and Nicotine Use Genomics and Sequencing Consortium (GSCAN) [13]. Additionally, a single study focusing on “problematic alcohol consumption” was included [35], utilizing data from Zhou et al.’s comprehensive GWAS published in Nature Neuroscience in 2020 [6]. This study conducted a total of 64 meta-analyses to investigate the health risks associated with drinking patterns. The analysis of results from all studies primarily utilized the IVW method, with the majority employing the Weighted Median and MR-Egger methods. Furthermore, several studies reported estimates for alcohol consumption after adjusting for smoking initiation.

### 3.3. Summary of the Relationship

Findings indicate that genetic predisposition to alcohol consumption, independent of smoking, was significantly correlated with a decreased risk of Parkinson’s disease (OR = 0.76 and 95%CI: 0.64–0.91), hyperplasia of the prostate (OR = 0.79 and 95%CI: 0.65–0.96), and rheumatoid arthritis (OR = 0.69 and 95%CI: 0.49–0.96). Conversely, it was associated with an increased risk of chronic pancreatitis (OR = 1.98 and 95%CI: 1.14–3.44), colorectal cancer (OR = 1.45 and 95%CI: 1.13–1.85), and head and neck cancers (OR = 1.90 and 95%CI: 1.10–3.28). Genetic predisposition to problematic alcohol use showed a strong association with an increased association with alcoholic liver disease (OR = 4.26 and 95%CI: 1.98–9.19), cirrhosis (OR = 2.76 and 95%CI: 1.22–6.22), acute pancreatitis (OR = 2.41 and 95%CI: 1.52–3.82), chronic pancreatitis (OR = 2.67 and 95%CI: 1.45–4.91), pneumonia (OR = 1.23 and 95%CI: 1.01–1.50), and head and neck cancer (OR = 2.20 and 95%CI: 1.03–4.73). And it was associated with a decreased risk of heart failure (OR = 0.74 and 95%CI: 0.56–0.99) (Figure 1; Appendix A).

### 3.4. Circulatory System Disease

Within the scope of the ten identified circulatory system diseases, genetic predisposition to alcohol consumption was associated with an elevated risk of both stroke (OR = 1.19 and 95%CI: 1.02–1.39) and hypertension (OR = 1.05 and 95%CI: 1.00–1.10), as detailed in Figure 2 and Appendix A. Genetic liability to problematic alcohol use was associated with a decreased risk of heart failure (Figure 2; Appendix A).

### 3.5. Digestive System Diseases

Genetic predisposition to alcohol consumption was associated with an increased risk of alcoholic liver disease (OR = 3.67 and 95%CI: 2.18–6.19), cirrhosis (OR = 2.40 and 95%CI: 1.48–3.89), duodenal ulcer (OR = 1.86 and 95%CI: 1.31–2.63), chronic pancreatitis (OR = 1.98 and 95%CI: 1.26–3.10), and chronic periodontitis (OR = 1.49 and 95%CI: 1.17–1.90), as shown in Figure 3 and Appendix A. Similarly, genetic susceptibility to problematic alcohol use was associated with an increased risk of alcoholic liver disease (OR = 4.26 and 95%CI: 1.98–9.19), cirrhosis (OR = 2.76 and 95%CI: 1.22–6.22), and both acute (OR = 2.41 and 95%CI: 1.52–3.82) and chronic pancreatitis (OR = 2.67 and 95%CI: 1.45–4.91), a finding detailed in Figure 3 and Appendix A.

### 3.6. Nervous System Diseases and Mental and Behavioral Disorders

Genetic predisposition to alcohol consumption was associated with increased risk of epilepsy (OR = 1.20 and 95%CI: 1.04–1.38), while divergent results were noted for Parkinson’s disease (OR = 0.79 and 95%CI: 0.67–0.93), as indicated in Figure 3 and outlined in Appendix A. In contrast, the investigations into problematic alcohol use did not yield any substantial or consistent evidence of an association, as depicted in Figure 4 and detailed in Appendix A.

### 3.7. Neoplasms

In the realm of neoplasms, genetic predisposition to alcohol consumption exhibited correlations with various cancers, including colorectal (OR = 1.33 and 95%CI: 1.06–1.67), esophageal (OR = 2.48 and 95%CI: 1.18 = 5.22), gastric (OR = 1.90 and 95%CI: 1.05–3.43), lung (OR = 1.68 and 95%CI: 1.28–2.22), oral and oropharyngeal (OR = 9.96 and 95%CI: 5.33–18.6), kidney (OR = 0.64 and 95%CI: 0.42–0.99), as well as head and neck cancers (OR = 1.85 and 95%CI: 1.19–2.89), with each of these showing an increased risk with the exposure phenotype. Notably, this pattern did not extend to kidney cancer, as detailed in Figure 5 and Appendix A. Conversely, genetic liability to problematic alcohol use was distinctly linked to the elevated risk of head and neck cancers (OR = 2.20 and 95%CI: 1.03–4.73) only, as indicated in Figure 5 and Appendix A.

### 3.8. Other Diseases

Genetic predisposition to alcohol consumption was linked to an elevated risk of spontaneous abortion (OR = 1.19 and 95%CI: 1.05–1.35) and severe COVID-19 (OR = 1.28 and 95%CI: 1.00–1.64) outcomes. Conversely, it was inversely associated with the risk of prostate hyperplasia (OR = 0.80 and 95%CI: 0.68–0.95) and rheumatoid arthritis (RA) (OR = 0.68 and 95%CI: 0.53–0.87), as detailed in Appendix A. Additionally, a genetic tendency toward problematic alcohol use was found to be correlated with an increased risk of pneumonia (OR = 1.23 and 95%CI: 1.01–1.50), as extensively described in Appendix A.

### 3.9. Sensitivity Analyses

The findings derived from the MR-Egger and WME methods largely align with the primary outcomes of the study. In addition to the established associations of genetic predisposition to problematic alcohol use with head and neck cancers (IVW: OR = 2.20; WME: OR = 2.43; MR-Egger: OR = 0.35) and pneumonia (IVW: OR = 1.23; WME: OR = 1.17; MR-Egger: OR = 0.57), an inverse relationship between genetic predisposition to alcohol consumption and spontaneous abortion (IVW: OR = 1.19; WME: OR = 1.13; MR-Egger: OR = 0.92) was noted in the MR-Egger analysis, as detailed in Appendix A.

Furthermore, even after adjusting for smoking initiation, the genetic predisposition toward alcohol consumption continued to show associations with Parkinson’s disease (OR = 0.76 and 95%CI: 0.64–0.91), chronic pancreatitis (OR = 1.98 and 95%CI: 1.14–3.44), benign prostatic hyperplasia (OR = 0.79 and 95%CI: 0.65–0.96), rheumatoid arthritis (OR = 0.69 and 95%CI: 0.49–0.96), colorectal cancer (OR = 1.45 and 95%CI: 1.13–1.85), and head and neck cancers (OR = 1.90 and 95%CI: 1.10–3.28). Notably, even following FDR correction, genetic predisposition toward alcohol consumption was still observed to be significantly associated with Parkinson’s disease and colorectal cancer, as indicated in Figure 1 and Appendix A.

Additionally, after applying FDR correction to the results on problematic alcohol use, significant associations persisted for genetic predisposition to problematic alcohol use in relation to alcoholic liver disease, acute pancreatitis, and chronic pancreatitis, as shown in Figure 1 and Appendix A.

## 4. Discussion

This study undertook a meta-analysis of 64 previously published MR studies, encompassing 151 de novo MR analyses of 76 distinct primary outcomes, aiming to investigate the association between genetic predisposition to drinking behaviors and the risk of various diseases. Our research conducted an extensive meta-analysis on two genetically inferred dimensions of alcohol use phenotypes, revealing causal relationships with 23 distinct diseases. These diseases include epilepsy, Parkinson’s disease, alcoholic liver disease, cirrhosis, duodenal ulcer, pancreatitis, chronic periodontitis, spontaneous miscarriage, benign prostatic hyperplasia, rheumatoid arthritis, pneumonia, severe COVID-19, heart failure, stroke, hypertension, as well as various cancers, such as colorectal, esophageal, gastric, lung, oral and oropharyngeal, kidney, and head and neck cancers. Interestingly, the two patterns of alcohol use demonstrated potential inverse correlations with certain conditions, including Parkinson’s disease, benign prostatic hyperplasia, rheumatoid arthritis, heart failure, and kidney cancer.

The World Health Organization (WHO) classifies alcohol as a contributing factor to more than 200 disease and injury conditions. These include mental and behavioral disorders, liver cirrhosis, specific cancers, and cardiovascular diseases. A comprehensive prospective study involving over 512,000 adults in China has elucidated the observational and genetic associations of alcohol consumption with a broad spectrum of disease outcomes, specifically in male participants. This cohort study found that alcohol consumption was associated with significantly increased risks for 61 diseases [46]. A cohort study in Korea involving 33,198 adults found that alcohol consumption was significantly and positively associated with an increased risk of cancer mortality in a dose-dependent manner, starting with light drinkers [47]. Our results were in broad agreement with these previous observational findings.

MR leverages genetic variants as instrumental variables (IVs) to estimate the causal effects of exposure factors on health outcomes. This approach uses the genetic association between specific genotypes and exposures to infer causality, effectively simulating a randomized controlled trial. For instance, in our study, IVs for genetic predisposition to alcohol consumption, such as the SNP rs1229984 in the ADH1B gene, can influence drinking behavior and modify the amount of alcohol consumed. These genetic links allow us to model the effect of alcohol consumption on various diseases, isolating the influence of alcohol intake from confounding factors typically present in observational studies. By employing these genetically determined proxies for alcohol consumption, MR provides a powerful tool for understanding how changes in alcohol intake can affect disease risk. In our de novo MR analysis investigating the relationship between genetically predicted alcohol consumption and alcoholic liver disease, the primary IVW method identified a significant association (OR = 3.67, 95% CI = 2.18–6.19). Similarly, the MR analysis of genetic predisposition to problematic alcohol use demonstrated a strong association with alcoholic liver disease, where the IVW method yielded an OR of 4.26 (95% CI = 1.98–9.19). In the de novo MR of alcohol consumption and cirrhosis, the IVW results indicated a significant relationship (OR = 2.45, 95% CI = 1.50–3.99). Moreover, our comprehensive meta-analysis, which integrated effect estimates from various data sources along with MR findings, suggested an increased risk of cirrhosis associated with genetic predisposition to alcohol consumption (IVW OR = 2.40, 95% CI = 1.48–3.89). For the association between alcohol consumption and chronic pancreatitis, the primary IVW analysis demonstrated a significant association (OR = 1.90, 95% CI = 1.18–3.05). This finding was further supported by a sensitivity analysis, which confirmed the robustness of these results. An extensive meta-analysis that integrated additional effect estimates further reinforced the increased risk of chronic pancreatitis associated with genetic predisposition to alcohol consumption. Even after MVMR adjustment, the link between genetically predicted alcohol consumption and chronic pancreatitis remained significant (OR = 1.98, 95% CI = 1.14–3.44). Sensitivity analyses further confirmed this robust association, demonstrating it was unaffected by genetic predispositions to smoking initiation.

Epidemiological research has uncovered a complex and non-linear relationship between patterns of alcohol consumption and circulatory system diseases [48]. The health implications of alcohol intake appear to depend on both the quantity consumed and the patterns of consumption. In this study, we explored the relationship between two patterns of alcohol use and ten types of circulatory system diseases. This was achieved using IVs for both genetic predisposition to alcohol consumption (“drinks per week”) and problematic alcohol use, combined with a comprehensive meta-analysis approach. Our results indicate that a genetic predisposition to alcohol consumption could increase the risk of hypertension and stroke. However, this association appears to diminish when adjustments are made for smoking as a confounding factor. Our findings paralleled those of the observational studies [49]. Interestingly, genetic predisposition to problematic alcohol use did not exhibit a statistically significant association with these two conditions. Conversely, the MR analysis results for problematic drinking suggested that excessive alcohol consumption might offer a protective factor against heart failure, while no such association was detected for genetic predisposition to alcohol consumption (“drinks per week”). Numerous population studies suggest that moderate drinking may benefit cardiovascular health by lowering the risk of heart failure. However, genetic epidemiological data do not support a causal relationship between alcohol consumption and the risk of heart failure [50,51]. A Korean study observed that light-to-moderate alcohol consumers exhibited a lower risk of heart failure compared to non-drinkers. However, increasing alcohol consumption from moderate to heavy levels was associated with a heightened risk of heart failure [51]. In our MR study, it was challenging to precisely quantify the specific amount of alcohol consumption directly associated with disease risk. Additionally, potential unidentified confounding factors could have influenced the outcomes. The relationship between varying levels of alcohol use and circulatory system diseases merits further investigation through longitudinal population studies. The findings of our study should be considered preliminary, providing a basis for more extensive future research in this field.

Alcohol consumption is sometimes considered a protective agent against autoimmune diseases, potentially due to the increase in systemic acetate levels induced by alcohol intake. This elevation in acetate levels may negatively affect the humoral immune response, potentially diminishing immune system functionality [52]. Nonetheless, the European Union’s lifestyle recommendations for preventing the progression of rheumatic and musculoskeletal diseases advise that individuals at risk for rheumatoid arthritis (RA) and gout may experience relapses following alcohol consumption beyond certain thresholds [53]. A prospective study in China among adults found that increased alcohol consumption was associated with a higher risk of rheumatic joint disease in women, although such an association was not observed in men [54]. The relationship between alcohol consumption and RA remains a topic of considerable debate in studies investigating it. Our study, employing MR methods, supports the hypothesis that alcohol consumption at moderate levels may serve as a protective factor against RA. This protective association persists even after adjusting for smoking as a confounding variable. However, a significant limitation of our study is its focus on European populations, restricting its demographic scope. Given the conclusions of these observational studies, further exploration is necessary to unravel the nuances of the association between different drinking patterns, genders, and ethnicities and RA.

A series of meta-analyses examining the link between alcohol consumption and Parkinson’s disease (PD) risk have been conducted, consistently revealing an inverse relationship [55,56]. Our investigation further substantiates this link, identifying a significant link between genetic predictors of alcohol consumption and a decreased PD risk, as evidenced by an OR of 0.79 and a 95% CI ranging from 0.67 to 0.93. This finding was similar to results from the National Institutes of Health-American Association of Retired Persons (NIH-AARP) Diet and Health Study, which involved 306,895 participants and suggested that low-to-moderate beer consumption correlated with a reduced PD risk [57]. Conversely, research using the NeuroEPIC4PD cohort reinforces previous findings from longitudinal studies, showing a significant association between alcohol consumption and PD risk [58]. The phenomenon of reverse causation in observational studies, often resulting from behavioral changes associated with the disease, could explain these inconsistent findings. For example, individuals with Parkinson’s disease (PD) may naturally reduce or cease their alcohol consumption in response to their symptoms. Mendelian randomization (MR) uses genetic variations as a form of natural “randomization” to facilitate the exploration of causal relationships between modifiable lifestyle factors and a spectrum of health outcomes [59].

There is limited epidemiological evidence of a relationship between alcohol consumption and the risk of renal cancer and benign prostatic hyperplasia [3,60,61,62]. However, contrasting research has pinpointed both smoking and alcohol consumption as significant contributors to renal cancer risk [63]. The association between alcohol and renal cancer remains controversial. Our study suggests that moderate alcohol consumption might potentially serve as a protective factor against renal cancer, but this association lost statistical significance upon adjusting for smoking, a known confounding variable. Moreover, our findings revealed no association between problematic drinking and renal cancer, offering insights for future population-based or mechanistic studies. Additionally, no connection was observed between problematic alcohol use and kidney cancer risk. These varied outcomes underscore the necessity for more extensive and detailed population-based or mechanistic studies to elucidate the complex interplay between alcohol consumption and kidney cancer risk.

This study leverages the most recent GWAS aggregated statistical data for de novo MR analysis and synthesizes them with findings from included studies in a comprehensive meta-analysis. This approach provides genetic evidence elucidating the health risks and benefits associated with alcohol consumption. MR analyses could avoid the bias of reverse causation and environmental confounding. To ensure effective bias reduction, an MR analysis adheres to three fundamental assumptions: the relevance assumption, the exclusion restriction assumption, and the independence assumption [10]. The relevance assumption requires a robust and consistent correlation between IVs and the exposure factor. In our de novo MR analysis, we meticulously selected SNPs based on GWAS correlation thresholds and F-statistics to minimize weak instrument bias. The independence assumption mandates that IVs must be unaffected by confounders that could influence the exposure–outcome relationship. To meet this requirement, our study included genetic predictors specifically for alcohol consumption and smoking initiation, which helped mitigate potential confounding effects. Moreover, by focusing exclusively on individuals of European ancestry, we aimed to prevent race-related biases. We also conducted rigorous linkage disequilibrium (LD) testing to eliminate SNPs in LD. Lastly, the exclusion restriction assumption dictates that IVs should influence health outcomes only through the exposure variable. Accordingly, our approach involved excluding any SNPs that had direct associations with the outcomes, ensuring a more accurate interpretation of the causal relationships.

To rigorously assess pleiotropy, we employed two advanced statistical tests for sensitivity analyses: MR-PRESSO’s global test [64] and MR-Egger’s intercept test [65]. These tests help identify and discard outlier SNPs potentially affected by pleiotropy. Additionally, our analysis incorporated two robust methods to further ensure the validity of our findings. The MR-Egger method [65] addresses potential pleiotropic effects in instrumental variables, while the WME [66] is utilized to accommodate any ineffective IVs. The concordance of effect estimates obtained from the IVW method [67] with those from the MR-Egger and WME methods substantiates the reliability of our conclusions. By integrating this rigorous analytical framework with data from various sources, we enhance the robustness and validity of our findings.

This research is distinguished by its application of both MR and meta-analysis techniques, providing solid evidence on the causal links between alcohol consumption and associated disease outcomes. This dual approach not only highlights the direct effects of alcohol intake but also systematically explores the links between genetic predispositions to alcohol consumption and problematic alcohol use and their health consequences. By examining these relationships from a genetic perspective, the study enhances our understanding of how different patterns of alcohol use influence various diseases, thereby contributing valuably to the field of alcohol-related health research.

However, this study is not without its notable limitations. While various methods were employed to adhere to the three core assumptions of MR, challenges such as potential biases related to pleiotropy, measurement errors in SNP-exposure associations, and the specificity of the genetic variants used remain. Additionally, other unknown confounding factors might still influence the exposure–outcome relationship. Furthermore, some of the statistically non-significant findings in our study could be attributed to the IVs explaining only a minor fraction of the phenotypic variation. These IVs might not fully capture alcohol exposure across the lifespan. Phenotypic alterations resulting from non-genetic influences are not addressed in this study, highlighting an area for future research to use more representative IVs for a deeper understanding of disease etiology. Another limitation is the study’s focus on the European population, which does not account for the substantial variations in drinking patterns observed across different regions and ethnic groups. For instance, East Asian populations often consume significant quantities of strong liquor and exhibit a higher prevalence of problematic drinking due to cultural influences [68]. This highlights the necessity for further investigation into the causal relationships between drinking behaviors and disease outcomes across different ethnic backgrounds. Additionally, the two phenotypes used in our study, “alcohol consumption” and “problematic alcohol use”, are based on aggregated statistical data and do not provide detailed insights into the specific quantities of alcohol consumed. Alcohol consumption is characterized as regular, non-pathological alcohol consumption, while problematic alcohol use encompasses pathological drinking patterns, including clinically diagnosed alcohol use disorder (AUD) and alcoholism. This distinction underscores a significant challenge for future research, requiring a more nuanced exploration of these consumption patterns.

## 5. Conclusions

Utilizing both MR and meta-analyses, this study methodically assesses the causal relationships between genetic predisposition to alcohol consumption, problematic alcohol use, and various disease outcomes. Our results indicate that alcohol consumption serves as a risk factor for various diseases, and within the same disease category, risks vary between daily alcohol intake levels and problematic drinking. This variation suggests that drinking patterns may significantly influence the health risks associated with alcohol. The dynamics between regular and problematic alcohol use and certain immune-related disorders, cardiovascular diseases, and various cancers are complex and exhibit non-linear patterns. Due to inherent methodological constraints and limitations in the available data, this study was unable to thoroughly explore these intricate relationships. Future research should incorporate newer genetic data to more accurately determine the causal connections between different patterns of alcohol use and diverse health outcomes, thereby enriching our understanding of these complex interactions.

## Figures and Tables

**Figure 1 nutrients-16-01517-f001:**
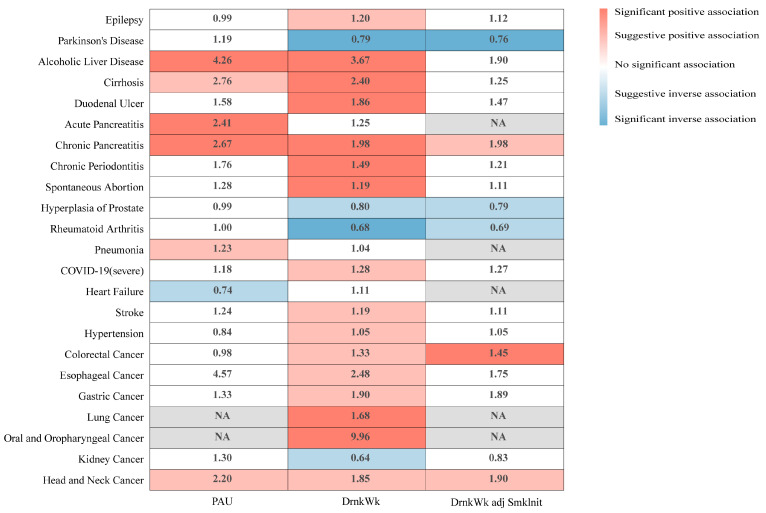
Meta-analytic summary of the relationship between alcohol consumption (measured in drinks per week) and problematic alcohol use and 23 diseases. Associations characterized by a raw *p*-value < 0.05 but a B-H-corrected *p*-value > 0.05 were deemed suggestive, whereas associations with a B-H-corrected *p*-value consistently < 0.05 were considered significant.

**Figure 2 nutrients-16-01517-f002:**
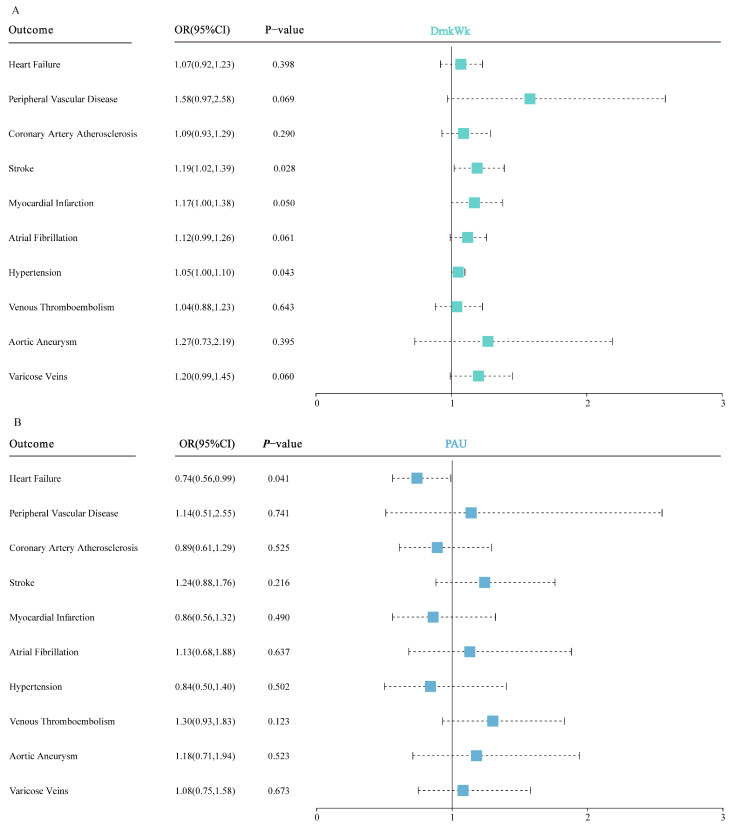
(**A**) Meta-analysis results of the association between genetic liability to alcohol consumption and the risk of circulatory system diseases. (**B**) Meta-analysis results of the association between genetic liability to problematic alcohol use and the risk of circulatory system diseases. The estimates represent odds ratios (ORs) with 95% confidence intervals (CIs) for one standard deviation increase in genetic liability to alcohol consumption and problematic alcohol use.

**Figure 3 nutrients-16-01517-f003:**
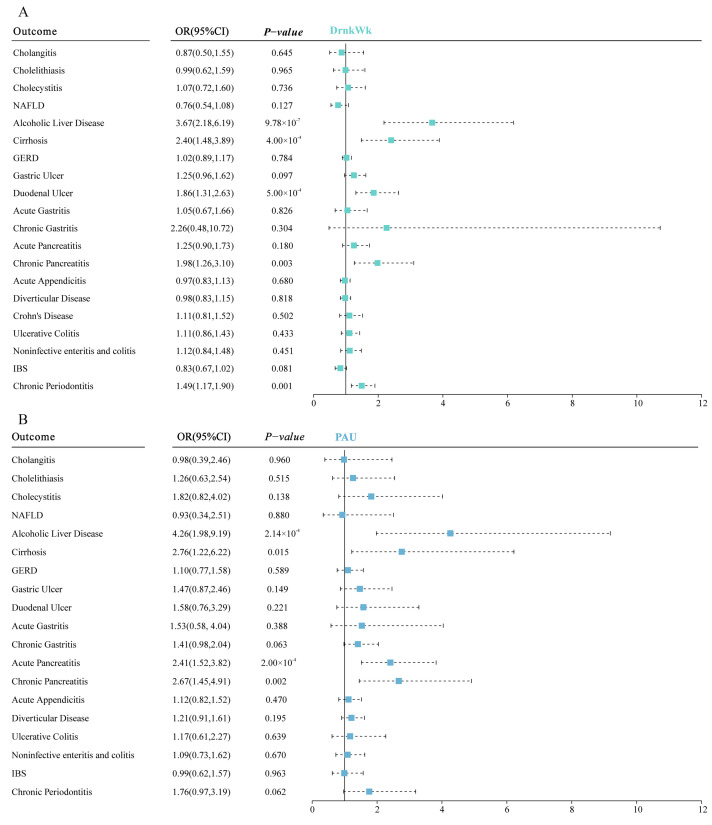
(**A**) Meta-analysis results of the association between genetic liability to alcohol consumption and the risk of digestive system diseases. (**B**) Meta-analysis results of the association between genetic liability to problematic alcohol use and the risk of digestive system diseases. The estimates represent odds ratios (ORs) with 95% confidence intervals (CIs) for one standard deviation increase in genetic liability to alcohol consumption and problematic alcohol use. NAFLD: nonalcoholic fatty liver disease; GERD: gastroesophageal reflux disease; IBS: irritable bowel syndrome.

**Figure 4 nutrients-16-01517-f004:**
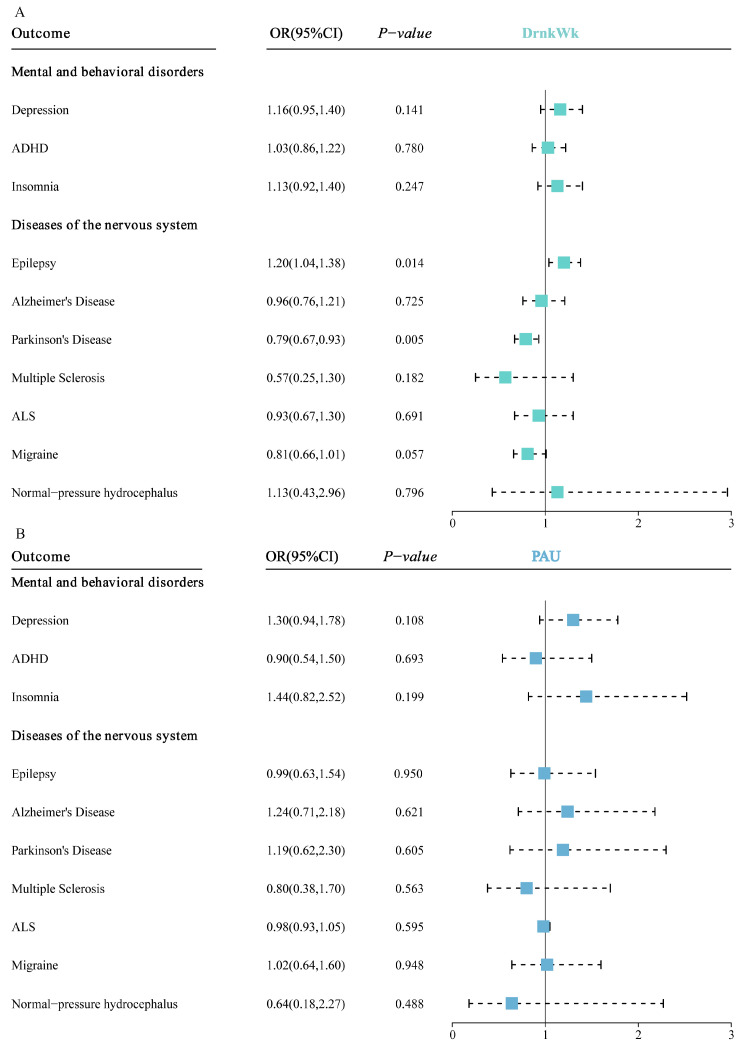
(**A**) Meta-analysis results of the association between genetic liability to alcohol consumption and the risk of nervous system diseases and mental and behavioral disorders. (**B**) Meta-analysis results of the association between genetic liability to problematic alcohol use and the risk of nervous system diseases and mental and behavioral disorders. The estimates represent odds ratios (ORs) with 95% CIs for one standard deviation increase in genetic liability to alcohol consumption and problematic alcohol use. ADHD: attention deficit hyperactivity disorder; ALS: amyotrophic lateral sclerosis.

**Figure 5 nutrients-16-01517-f005:**
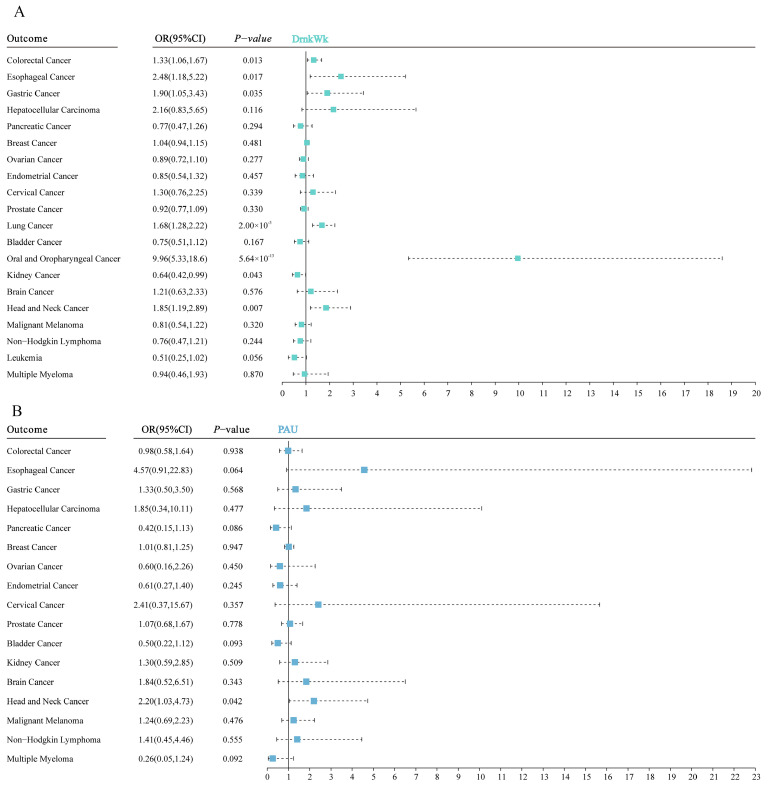
(**A**) Meta-analysis results of the association between genetic liability to alcohol consumption and the risk of neoplasms. (**B**) Meta-analysis results of the association between genetic liability to problematic alcohol use and the risk of neoplasms. The estimates represent odds ratios (ORs) with 95% confidence intervals (CIs) for one standard deviation increase in genetic liability to alcohol consumption and problematic alcohol use.

## Data Availability

GWAS summary data on “drink per week”, “smoking initiation”, and “problem alcohol use” were downloaded from GSCAN: “https://genome.psych.umn.edu/index.php/GSCAN (accessed on 15 June 2023)” and dbGaP: “https://www.ncbi.nlm.nih.gov/projects/gap/cgi-bin/study.cgi?study_id=phs001672.v3.p1 (accessed on 03 September 2023)”. GWAS summary data on “COVID-19” and “ALS” were downloaded from COVID-19 hg: “https://www.covid19hg.org/results/r7/ (accessed on 19 October 2023)” and the IEU OpenGWAS project: “https://gwas.mrcieu.ac.uk/ (accessed on 18 September 2023)”. For other diseases, GWAS summary data were selected from FinnGen(R9): “https://www.finngen.fi/en (accessed on 27 June 2023)”.

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
