# Peer review of "Alcohol Exposure and Disease Associations: A Mendelian Randomization and Meta-Analysis on Weekly Consumption and Problematic Drinking"

_nutrients, 2024, doi:10.3390/nu16101517_

Round 1
Reviewer 1 Report
Comments and Suggestions for Authors
This is a meta-analysis of studies describing possible association between alcohol consumption and diseases. The study is well presented and the findings are clearly presented.
Several points:
1. Please add a table specifying what is considered light, moderate and heave alcohol consumption (e.g. in glasses/week or day).
2. A comprehensive explanation should be added regarding genetic disposition to alcohol consumption as well as genetic disposition to problematic drinking. It is not clear what the authors mean by that term. Is it SNP associated consumption? What SNPs are the ones that are relevant?
3. Lines 274-281 are a repetition of the paragraph above it….
Comments on the Quality of English Languageseveral minor mistakes should be corrected.
Reviewer 2 Report
Comments and Suggestions for Authors
the authors adress the question of a so-called dual effect of alcohol use: moderate alcohol use supposely having health benefits and heavy alcohol use increasing health damages.
The authors apply a meta-analysis of published GWAS predicting various types of disease, select a set of SNPs associated with alcohol use or problematic alcohol use from two GWASes and test through Mendelian Randomization if the genetic risk of alcohol use and alcohol problematic use overlap with the genetic risk of those various types of diseases, most often by increasing the risk.
the MR methods applied here follow the good practices of this method.
but the research question itself does not reflect current scientific knowledge. the existence of heakth risks associated with even low alcohol use is already demonstrated: see PMID: 28220587, PMID: 19560604, and for specific health risks: high bood pressure, PMID: 34762198, cancer: PMID: 33633295, .....
futhermore, the MR approach may be inadequate to explore alcohol exposure over the lifespan because the genetic risk of the two phenotypes "alcohol use" or "problematic alcohol use" are defined at the time of the interview.
the authors should rephrase the title, abstract, and all introduction to avoid sensationalism in writing and describe facts.
the discussion should laso be more cautious regarding the impact of MR to demonstrate what is already known from epdemiological studies.
Round 2
Reviewer 2 Report
Comments and Suggestions for Authors
The authors adequatly adressed my previous comments and queries.
They modified the title, abstract, introduction, methods and discussion/conclusion.
The article has been muche imporved by those changes.